# UMMAN: Unsupervised Multi-graph Merge Adversarial Network for Disease Prediction Based on Intestinal Flora

## Abstract

The abundance of intestinal flora is closely related to human diseases, but diseases are not caused by a single gut microbe, but by a combination of a large number of microbial information. It exists a multiplex and implicit connection between gut microbes and hosts, which poses a great challenge to disease prediction through abundance information of gut microbes. Recently, several solution methods have been proposed and have shown the potential of predicting the corresponding diseases. However, these methods have difficulty in learning the inner association between gut microbes and hosts, resulting in unsatisfactory performance. This paper presents a novel architecture, **U**nsupervised **M**ulti-graph **M**erge **A**dversarial **N**etwork (UMMAN). UMMAN can obtain the embeddings of nodes in the Multi-Graph under unsupervised situation, so that it helps learn the multiplex relationship. At the first time, our method combines Graph Neural Network with intestinal flora disease prediction task. We use multiplex relation-types to construct the Original-Graph and destroy the relationship among nodes to get corresponding Shuffled-Graph. We introduce the Node Feature Global Integration (NFGI) module to represent the global features of the graph. Furthermore, we design a joint loss consists of adversarial loss and hybrid attention loss to make real graph embedding agree with the Original-Graph as much as possible and disagree with the Shuffled-Graph as much as possible. Comprehensive experiments on five classical OTU gut microbiome datasets demonstrate the effectiveness and stability of our method. (We will release our code soon.)

## 1 INTRODUCTION

According to incomplete statistics, there are at least 500 trillion microorganisms in the human body, which is more than 10 times the number of human cells(Qin et al., 2010). Microorganisms are not only large in number, but also of various types, including viruses, bacteria, fungi, eukaryotes, etc. All of these microbes together make up the human microbiome, an important area of modern biomedical research.

Some specific diseases are closely related to the abundance of microbial communities in the human body, and more than 80% of the microorganisms in the human body are lodged in the intestinal tract. As the largest microbial host place in the human body, the intestinal tract the microbiome in the gut has a direct impact on human health and diseases. For example, the researchers found that the proportion of Bacteroides and Actinobacteria in the intestines of obese and lean people was significantly different, and the proportion of Firmicutes also had an impact on obesity(Dicksved et al., 2008; Ley et al., 2006). Various metabolites of gut microbes, such as short-chain fatty acid and aromatic amino acids, all directly or indirectly induce type 2 diabetes(Zhu & Goodarzi, 2020; Qin et al., 2012). Intestinal microbes in patients with inflammatory bowel disease (IBD) are relatively disordered(Frank et al., 2007; Manichanh et al., 2006), and intestinal microbial metabolites such as acetate and butyrate play an important role in immune regulation. The concentration of butyrate decreased in patients with ulcerative colitis, which also revealed the influence of gut microbes on IBD. In recent years, with the rise of 16SrRNA sequencing technology(Liu et al., 2020), new studies have shown that the number of lactic acid bacteria in the intestines of patients with irritable bowel syndrome decreased significantly, while the number of Veillonella in patients with constipation-

predominant irritable bowel syndrome increased(Tana et al., 2010), which further confirmed the intestinal has close connection between microbes and human health(Clemente et al., 2012; Song et al., 2021).

With the rapid improvement of medical technology, obtaining relevant data has become more convenient. The abundance of human intestinal flora can be obtained by using 16SrRNA sequencing technology(Dethlefsen et al., 2008), which were previously unobtainable by traditional means. Concomitantly, how to process the data to obtain the results of disease analysis poses a challenge to us. With the soaring cost of expert analysis and the bottleneck of traditional algorithms, utilizing artificial intelligence to solve such problems has become a rigid need.

However, the diagnosis of sample diseases based on the information of intestinal flora has not achieved satisfactory results, because it exists a close but multiplex connection between the intestinal flora and the hosts which strongly affect the diseases absolutely. The connection between the gut microbiota and the hosts is intricate, and it is difficult to accurately obtain the association in previous methods. Most existing methods only consider a single relation-type even ignore the connection. Graph learning(Xia et al., 2021) is good at dealing with graph-structured data with rich relationships (Tena Cucala et al., 2022; Cheng et al., 2023), and can represent information nearby at any depth. Therefore, we use Graph Neural Network to learn the connection between microorganisms and hosts, which effectively guides disease prediction. It is worth noting that our method does not depend on the label of the node in the process of obtaining the embeddings of the nodes and the graphs, which can obtain a satisfactory embedding in an unsupervised situation.

We summarize the contributions of this paper as follows:

- We are the first to introduce graph machine learning to the field of gut microbiota disease prediction. We propose a novel Unsupervised Multi-graph Merge Adversarial Network (UMMAN). It learns the multiplex connection between intestinal flora and hosts to guide the prediction of diseases.

- We use a variety of relation-types to build a Multi-graph, and shuffle the nodes of the corresponding graph to destroy the association among nodes. We introduce the adversarial loss and the hybrid attention loss as a joint loss in order to make the true embeddings agree with the Original-Graph and disagree with the Shuffled-Graph as much as possible.

- We propose Node Feature Global Integration (NFGI) Descriptor to represent the global embedding of a graph.

- Experiments on the benchmark datasets indicate that our UMMAN achieves state-of-the-art on the disease prediction task of gut microbiota, and also prove that our method is more stable than previous methods.

## 2 RELATED WORK

Benefiting from the rapid advancement of medical technology, many studies have been able to use OTU data for analysis (Kim et al., 2020; Edgar, 2013; McDowell et al., 2022). Pasolli et al. (2016) comprehensively evaluated the prediction tasks based on shotgun metagenomics and the method of microbial phenotype association evaluation, mainly using support vector machines and random forest models to predict diseases, and with the help of Lasso and elastic net (ENet) Regularized Multiple Logistic Regression. In their study, cirrhosis was the most predictive disease, and the model used in the study had good generalization ability for cross-stage data, but poor generalization ability for cross-datasets. Sharma et al. (2020)proposed TaxoNN to predict the link between gut microbes and diseases, but only used two datasets to test the effect.

Manandhar et al. (2021) used fecal 16S metagenomics data to analyze 729 IBD patients and 700 healthy individuals through 5 machine learning methods. After identifying 50 microorganisms with significant differences, the prediction was obtained through the random forest algorithm. Based on the data of the gut project in the United States, Liñares Blanco et al. (2021)used the glmnet model and the random forest model to predict the country of origin. Wong et al. (2021)investigated the possible gastrointestinal effects of neratinib in the treatment of breast cancer. By collecting stool samples from 11 drug-taking patients and classifying patients who may develop diarrhea by a tree-based classification method.

In general, at present, for data in the form of OTU datasets that are extended to table types, the traditional machine learning algorithm may be relatively more effective (Zhou & Gallins, 2019). However, these algorithms are relatively fixed and there is not much space for improvement, reaching a bottleneck on the problem. Apart from that, basic CNNs perform well in many projects (Perez & Wang, 2017), but it can't beat the traditional machine learning algorithm in this case. This is because the basic CNN's kernel cannot learn arbitrary transformed features, which is not suitable for OTU datasets. The exchange of rows and columns of the datasets do not affect the results of pattern recognition and the connection between the intestinal flora and the hosts is intricate, which poses a great challenge to performance improvement. We propose the UMMAN unsupervised method, combining Graph Neural Network with gut microbiome for the first time, and it has achieved excellent performance on benchmark datasets.

## 3    METHOD

In this section, we will first introduce the overview of the architecture of our method — UMMAN. In the following section, we introduce how to construct the Original-Graph and the Shuffled-Graph. We then present the details of the two stages of the Node Feature Global Integration descriptor: node-level stage and graph-level stage. Then we introduce the attention block to extract the embedding of each node. We also state the joint loss which consists of adversarial loss $\mathcal{L}_{adv}$ and hybrid attention loss $\mathcal{L}_{h-attn}$ in the end.

### 3.1    THE OVERVIEW OF UMMAN ARCHITECTURE

The architecture of UMMAN we propose will be introduced detailedly in this part. As stated above, it exists close but extremely complex connection between gut microbes and hosts, and such relation is implicit. Specifically, it is not the abundance of a single intestinal microbe that can establish a direct relationship with the final disease, but it is caused by the combined information of various microbes. For tabular datasets in the form of OTUs, arbitrarily swapping rows or columns will not affect the classification results. Therefore, how to learn the connection between gut microbes and hosts has become the biggest difficulty in this field, and we are the first to combine Graph Learning with gut flora disease prediction tasks which helps learn the connection. The following will introduce the architecture of UMMAN that we propose in detail.

The overview architecture of UMMAN is shown in Figure 1. We use multiplex indicators to measure the similarity among nodes to build the Original-Graph. In order to make UMMAN learn associations more accurately, we destroy the Original-Graph's association among nodes to get Shuffled-Graph. The Original-Graph and Shuffled-Graph are updated through the Graph Convolutional Network at the same time in order to obtain the embeddings of each node. We introduce the attention block to obtain the embeddings of the nodes in the Original-Graph and the embeddings of the nodes in the Shuffled-Graph respectively. In addition, we design a Node Feature Global Integration (NFGI) descriptor to denote the embedding of a graph. Then, in order to make the true embedding that includes the complex and implicit relationship between gut microbes and hosts agree with the Original-Graph as much as possible and disagree with the Shuffled-Graph as much as possible, we design the total joint loss function consists of an adversarial loss and hybrid attention loss.

### 3.2    CONSTRUCT MULTI-GRAPH AND NODE EMBEDDING

We find that the abundance data of a part of the intestinal flora in the dataset was 0 in most hosts whose impact on the performance of the algorithm is detailed above. If all the data of the original dataset is retained, it may cause some intestinal flora to be almost completely absent in all hosts. We can consider such data as "dirty data", which will lead to training effects decline. Therefore, we delete the rows whose abundance values are all 0 and the rows with too many 0s in the abundance value to improve the effect of feature extraction.

In order to make the motivation more convincing, we show the relationship of the 10 most abundant flora in the cirrhosis dataset in the sample intestine in Figure 2. The diagonal graph describes the distribution histogram of the flora, and the off-diagonal represents the relationship between the flora and the abundance in the intestinal tract of the sample, where the red dots represent healthy hosts, and the blue dots represent diseased samples. It can be found that the correlation is not obvious and

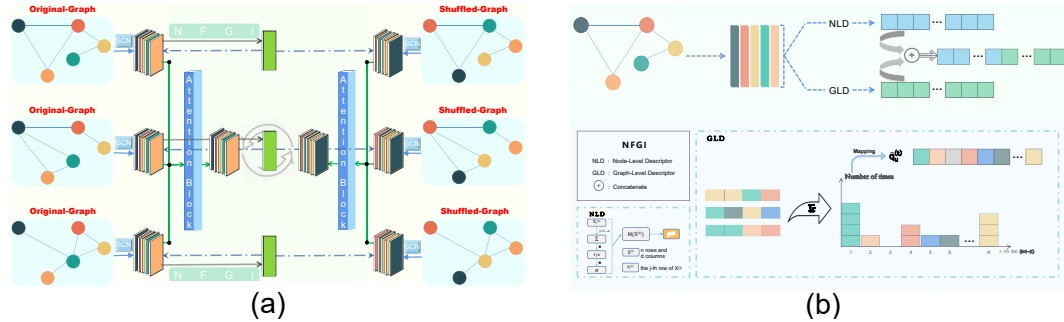

(a)                           (b)

Figure 1: The architecture of the UMMAN we propose. The subgraph on the left is the core architecture of the entire network. The subgraph on the right is the specific process of the NFGI module, including Node-level stage and Graph-level stage.

appears to be disorganized, which proves that it is unreliable to measure the relation-type between hosts only based on the abundance of flora. So we use multiplex relation-types (Beals, 1984; Jurman et al., 2009) to measure the relationship between vectors during the initialization of building the edges among nodes in the graph:

$$\mathcal{S}_1(m, n) = \frac{\sum_{i=1}^{d} |m_i - n_i|}{\sum_{i=1}^{d} m_i + \sum_{i=1}^{d} n_i} \tag{1}$$

$$\mathcal{S}_2(m, n) = \sqrt{\sum_{i=1}^{d} (m_i - n_i)^2} \tag{2}$$

$$\mathcal{S}_3(m, n) = \sum_{i=1}^{d} \frac{|m_i - n_i|}{|m_i| + |n_i|} \tag{3}$$

where Bray Curtis Distance, Euclidean Distance and Canberra Distance are used to initialize a graph. Two nodes are considered to be connected if the distance between their embeddings is below a variable threshold $\theta$, so we get the Original-Graph. In order to make our model more robust, and to obtain the correlation among nodes more reliably, we design the adversarial control group, i.e., keep the position of the edges unchanged, randomly disrupt the nodes to train part of the discriminator at the same time get the adversarial loss, used to backpropagate to update the parameters. The specific process will be introduced in detail later.

We introduce an encoder module for each relation type inspired by the Graph Convolutional Network(Kipf & Welling, 2016), aiming to obtain the embedding of each node in the graph. We define a conditional function $\mathcal{F}$ as an update function between layers, the Shuffled-Graph is operated by the same process $\mathbb{R}^{N \times F} \hookrightarrow \mathbb{R}^{N \times D}$:

$$\mathcal{X}_j^{(l+1)} = \mathcal{F}(\mathcal{X}_j^{(l)}, \hat{A} | W^{(l)}, b^{(l)}) = \sigma \left( \sum_{j \in N_i} \hat{\mathcal{D}}^{-\frac{1}{2}} \hat{\mathcal{A}} \hat{\mathcal{D}}^{-\frac{1}{2}} \mathcal{X}_j^{(l)} W^{(l)} + b^{(l)} \right) \tag{4}$$

where $\mathcal{X}_j^{(l)}$, is the embedding of the l layer of the node whose index is j, $N_i$ represents the set of nodes adjacent to $\mathcal{X}_j^{(l)}$, $\hat{A}$ and $\hat{\mathcal{D}}$ are the adjacency matrix and degree matrix of a certain graph, $W^{(l)}$ and $b^{(l)}$ are the trainable parameters of the l-th layer's weight matrix and bias, $\sigma$ is the nonlinearity layer which is designed as ReLU in our method. After being updated by Graph Convolutional Network, $X_j^{(t)}$ is a D-dimensional tensor representing the embedding of the node with index j of the t-th relation-type.

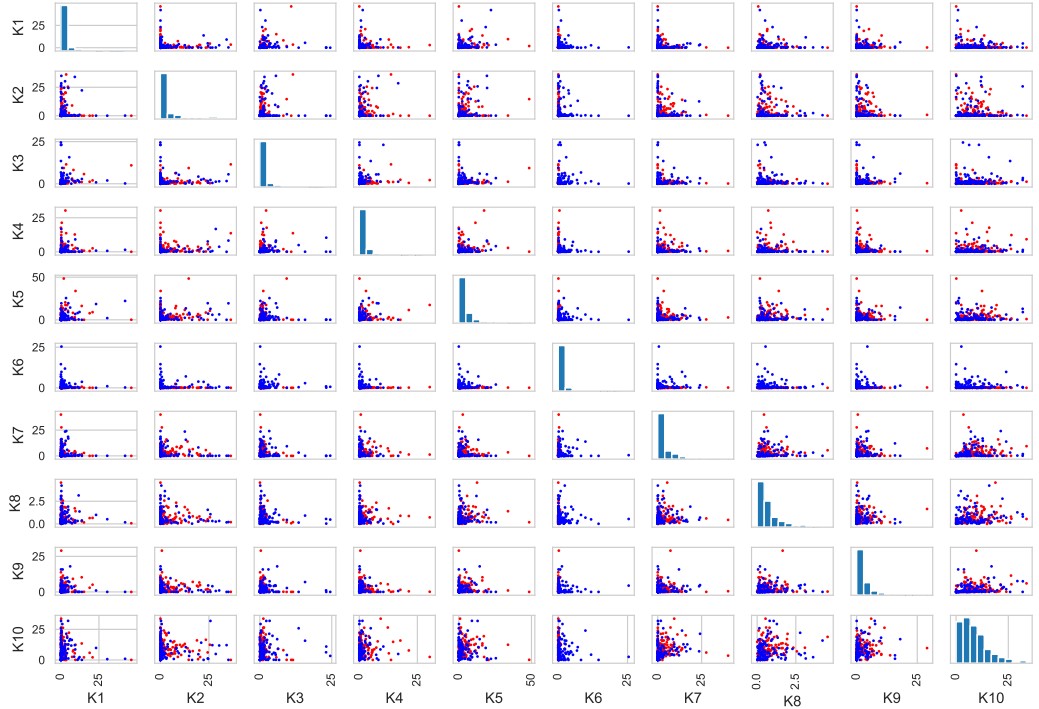

Figure 2: Abundance correlations of the top 10 most abundant gut microbes in the Cirrhosis dataset.

### 3.3 NODE FEATURE GLOBAL INTEGRATION DESCRIPTOR WITH TWO STAGES

In the NFGI module, we propose a novel contribution to characterize a graph $\mathcal{J}^{(t)}/\tilde{\mathcal{J}}^{(t)}$ with two stages($\mathcal{J}^{(t)}$ is the Original Graph constructed according to the t-th relation-type and $\tilde{\mathcal{J}}^{(t)}$, is the corresponding graph after being shuffled, since the operations of the two graphs are equivalent, we use $\mathcal{J}^{(t)}$ uniformly in the following text). In the conventional method, only the features of each node are fused, but using this method to characterize a graph only considers the information of each node. Therefore, our NFGI module adopt a two-stage process, i.e., node-level stage and graph-level stage to jointly characterize the features of the graph to better obtain the embedding.

***Node-Level stage***. We first compute the node-level contribution to graph $\mathcal{J}^{(t)}$ based on the characteristics of each node. That is, to fuse the embedding of each node after the Graph Convolutional Network. We employ a function $\mathcal{M} : \mathbb{R}^{N \times D} \hookrightarrow \mathbb{R}^{D}$:

$$p^{(t)} = \mathcal{M}(X^{(t)}) = \sigma(\frac{1}{N} \sum_{i=1}^{N} X_i^{(t)}) \tag{5}$$

$p^{(t)}$ summarizes the node-level features of the graph $\mathcal{J}^{(t)}$. Among them, $X^{(t)}$ is a matrix with n rows and d columns, $X_i^{(t)}$ is the i-th row of matrix $X^{(t)}$ and $\sigma$ denotes the logistic sigmoid nonlinearity function.

***Graph-Level stage***. In the graph-level stage, we concatenate the embedding of each node to build a tensor, and then flatten the tensor to get the total information of the whole graph which can form K bins. Let $\{x_j^*\}_{j=1\dots n}$ be the each center of the histogram bins. The value included in each bin is within the range of r centered on $\{x_j^*\}$. Given a certain feature value $x_i$ , we define a function $\Psi$, , the index $\Psi(x_i^{(t)})$ corresponding to the bin that the value belongs to. On the wholethe function $\mathcal{G}$ denotes the embedding of the graph in the graph-level stage:$\mathbb{R}^{N \times D} \hookrightarrow \mathbb{R}^{K}$:

$$\tilde{q}_u^{(t)} = \mathcal{G}(X^{(t)}) = \sum_{i=1}^{N \times D} C(|x_i^{(t)}|)\delta[\Psi(x_i^{(t)}) - u] \qquad (6)$$

$$q^{(t)} = \mathcal{E}[\tilde{q}_{u=1\ldots K}^{(t)}] \qquad (7)$$

where $\delta$ is the Kronecker delta function. The normalization constant C is derived by imposing $\sum_{i=1}^{N \times D} \tilde{x}_i^{(t)} = 1$ , performing data smoothing on the graph-level results. The graph-level embedding result can describe the global feature of the graph. Therefore, after concatenating each graph-level stage feature using the symbol $\mathcal{E}$ , the final graph-level embedding result is obtained: $\mathbb{R}^{N \times D} \hookrightarrow \mathbb{R}^{D+K}$. Finally, we concatenate the node-level embedding and graph-level embedding by the axis 0.

$$G^{(t)} = \mathcal{E}[\tilde{p}^{(t)}, \tilde{q}^{(t)}] \qquad (8)$$

## 3.4 MERGE EMBEDDING OF MULTI-GRAPH

After using Graph Convolutional Network to update the Original-graph and Shuffled-graphs, we get the embedding of each node. Each node in the Multi-graph $\mathcal{J}^{(t)}$ gets a feature vector to represent, and the attention block can update the embeddings of the corresponding nodes (Vaswani et al., 2017) of the Multi-graph to an embedding result representing the node feature. The attention block is defined as follows:

$$\boldsymbol{X}_{attn} = \boldsymbol{Attn}(X_i^{(t)} \mid t \in \mathcal{T}) = \sum_{t \in T} \frac{exp(query^{(t)} X_i^{(t)})}{\sum\limits_{t \in T'} exp(query^{(t')} X_i^{(t')})} X_i^{(t)} \qquad (9)$$

where $query^{(t)} \in \mathbb{R}^{D+K}$ is the feature vector of relation-type t. The Original Graphs and the Shuffled Graphs can obtain the embedding results $\boldsymbol{x}_i$ and $\tilde{\boldsymbol{x}}_i$ of each node after passing through the attention block. Finally, the embedding of nodes can be sent to the MLP for prediction.

## 3.5 LOSS FUNCTIONS

The loss functions of our model consists of an adversarial loss $\mathcal{L}_{adv}$ and hybrid attention loss $\mathcal{L}_{h-attn}$. The total loss function is as follows:

$$\mathcal{L} = \mathcal{L}_{adv} + \dot{\eta}\mathcal{L}_{h-attn} \qquad (10)$$

where $\dot{\eta}$ is a learnable coefficient for the loss terms.

### 3.5.1 ADVERSARIAL LOSS

We not only use a variety of relation-types to construct Multi-graph and obtain the embedding of each Node through Graph Convolutional Network and Attention Block, but also the Original-Grpah Randomly shuffle to break the correlation among nodes. Therefore, we calculate the positive correlation loss between the global embedding obtained by NFGI (Node Feature Global Integration Descriptor) and the embedding of the nodes of the Original-Graph obtained by each relation-type. At the same time, on the converse side, we calculate the negative correlation loss between the global embedding and the embedding of the nodes of the Shuffled-Graph, and define this joint loss as Adversarial Loss inspired by Pajot et al. (2018). In other words, we calculate the adversarial loss from the embedding obtained from the constructed Original-Graph (Positive) and the Shuffled-Graph (Negative) that destroys the correlation among nodes, which is defined as:

$$\mathcal{L}_{adv} = \sum_{t \in \mathcal{T}} \sum_{i=1}^{N} log\sigma((H^{(t)})^{\mathrm{T}} W^{(t)} X_i^{(t)}) + \sum_{t \in \mathcal{T}} \sum_{j=1}^{N} log(1 - \sigma((H^{(t)})^{\mathrm{T}} \tilde{W}^{(t)} \tilde{X}_i^{(t)})) \quad (11)$$

### 3.5.2 HYBRID ATTENTION LOSS

The Hybrid Attention Loss we proposed comprehensively considers the embedding after the nodes of the Original-Graph and the Shuffled-Graph pass through the attention block. The Hybrid Attention Loss $\mathcal{L}_{h-attn}$ can make the global embedding matrix of real graph agree with $\boldsymbol{X}_{attn}^{(t)}$, and disagree with $\tilde{\boldsymbol{X}}_{attn}^{(t)}$, thereby improving the confidence of the attention block. The Hybrid Attention Loss function is defined as:

$$\mathcal{L}_{h-attn} = (\mathcal{P} - \boldsymbol{X}_{attn})^2 - (\mathcal{P} - \tilde{\boldsymbol{X}}_{attn})^2 \quad (12)$$

## 4 EXPERIMENT

### 4.1 DATASETS

Five classical gut microbiota datasets are used for the experiments. We use five available disease-associated metage nomic datasets spanning four diseases: liver cirrhosis, inflammatory bowel diseases (IBD), obesity, and type 2 diabetes in Asia and Europe, respectively. These datasets have recorded the abundance of 1331 gut microbes in the sample intestine, as well as information such as gender, age, and region.

***Cirrhosis*** (Qin et al., 2014). Qin et al. extracted total DNA libraries from fecal samples of patients with cirrhosis and healthy controls, which can characterize the gut microbiome of patients with cirrhosis. Specifically, they used Illumina HiSeq 2000 for sequencing, which produced an average of 4.74 Gb of high-quality sequences per sample, and a total of 860 Gb of 16SrRNA gene sequence data.

***IBD*** (Qin et al., 2010). Qin et al. (2010) collected stool samples from volunteers and performed Illumina GA sequencing. All Reads were assembled using Soapnovo19. Using BLAT36 to construct a non-redundant gene set for inflammatory bowel disease by pairwise comparison of all genes. Each sample yields an average of 4.5Gb of high-quality sequences.

***Obesity*** (Le Chatelier et al., 2013). Obesity is one of the most serious (proportional) gut microbiota-related diseases facing the world. Le et al. extracted the abundance of more than a thousand species of flora in the gut by sequencing fecal samples from Obesity patients and thinner stature.

***T2D*** (Qin et al., 2012). In order to obtain the DNA data of the flora in the samples, Qin et al. (2010) collected stool samples from patients with type 2 diabetes and healthy controls, using the whole genome sequencing method, and then sequenced all the DNA samples, and analyzed the intestinal microbial DNA of 345 Chinese Two-stage MGWAS was performed for deep shotgun sequencing, with an average of 2.61 Gb per sample and a total of 378.4 Gb of high-quality DNA data.

***WT2D*** (Karlsson et al., 2013). Shotgun sequencing is used to analyze the whole genome sequence of fecal samples from European women, sequenced on Illumina HiSeq 2000, obtained an average of 2Gb of sequencing data per sample, and a total of about 449 Gb of data. Different from T2D, the target of T2D dataset is Chinese, while WT2D is sampled in Europe.

### 4.2 COMPARISON WITH EXISTING WORK

We carry out quantitative comparisons on the results between our method and existing work on five datasets. The Random Forest and Support Vector Machine can be used in the disease prediction of intestinal flora (Pasolli et al., 2016), although the results obtained are acceptable, the performance of such algorithms has reached the bottleneck so that it is difficult to improve. For the methods of the Convolutional Neural Network series, since the convolution kernel is better at extracting closely arranged features, but this task is in the form of a tabular dataset, any exchange of specific rows and columns will not affect the disease of the sample. Therefore, the connection between the flora

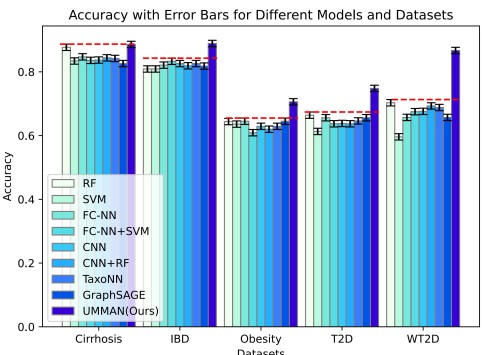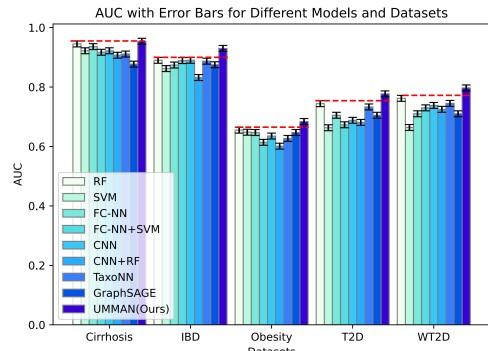

Figure 3: Intuitive comparison of our method with previous work on the five OTU datasets.

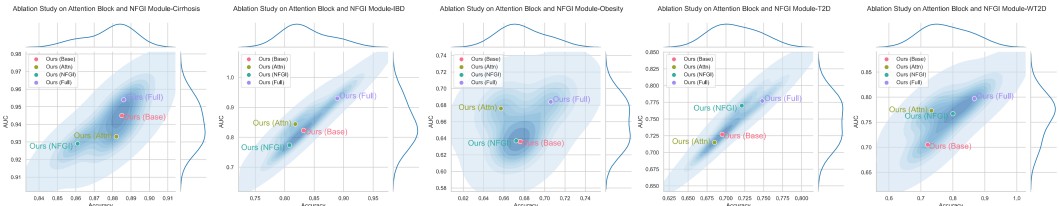

Figure 4: The comparison on four variants of our method.

learned by the convolution kernel is unreliable. We innovatively combined a graph machine learning algorithm with the gut microbiota disease prediction task. We first compared the performance results of our method and the previous 8 methods on the Acc and AUC indicators on the 5 datasets, as shown in Table 1 (bold indicates the best performance, underline indicates the second best performance).

Table 1: Comparison of our method with previous methods on the Accuracy and the Area Under Curve metrics in the Cirrhosis, IBD, Obesity, T2D and WT2D datasets.

| Method | Cirrhosis | | IBD | | Obesity | | T2D | | WT2D | |
|---|---|---|---|---|---|---|---|---|---|---|
| | Acc | AUC | Acc | AUC | Acc | AUC | Acc | AUC | Acc | AUC |
| RF (Pasolli et al., 2016) | 0.877 | 0.945 | 0.809 | 0.890 | 0.644 | 0.655 | 0.664 | 0.744 | 0.703 | 0.762 |
| SVM (Pasolli et al., 2016) | 0.834 | 0.922 | 0.809 | 0.862 | 0.636 | 0.648 | 0.613 | 0.663 | 0.596 | 0.664 |
| FC-NN (Rumelhart et al., 1986) | 0.847 | 0.936 | 0.821 | 0.874 | 0.645 | 0.647 | 0.656 | 0.705 | 0.657 | 0.71 |
| FC-NN+SVM (Rumelhart et al., 1986; Pasolli et al., 2016) | 0.836 | 0.917 | 0.833 | 0.889 | 0.609 | 0.614 | 0.637 | 0.673 | 0.675 | 0.73 |
| CNN (Krizhevsky et al., 2012) | 0.837 | 0.922 | 0.826 | 0.890 | 0.629 | 0.635 | 0.638 | 0.688 | 0.676 | 0.738 |
| CNN+RF (Krizhevsky et al., 2012; Breiman, 2001) | 0.844 | 0.907 | 0.819 | 0.832 | 0.62 | 0.601 | 0.637 | 0.681 | 0.693 | 0.725 |
| TaxoNN (Sharma et al., 2020) | 0.842 | 0.911 | 0.826 | 0.887 | 0.629 | 0.627 | 0.646 | 0.733 | 0.688 | 0.745 |
| GraphSAGE (Hamilton et al., 2017) | 0.826 | 0.877 | 0.818 | 0.875 | 0.645 | 0.647 | 0.656 | 0.705 | 0.657 | 0.71 |
| UMMAN(Ours) | **0.886** | **0.954** | **0.889** | **0.930** | **0.706** | **0.684** | **0.748** | **0.777** | **0.867** | **0.797** |

Our method, UMMAN, achieves the best performance on the metrics across the five datasets on all indicators. From the experimental results in Table 1 and the visual comparison of Figure 3, it can be found that almost all the optimal indicators and suboptimal indicators are derived from our method and Random Forest. This may be due to the robustness of traditional machine learning algorithms. Therefore, in order to prove that our method (UMMAN) not only achieves SOTA on the result value, but also surpasses the previous methods in all aspects in terms of stability and other indicators. Table 2 compares the performance of our method and traditional machine learning algorithms on five datasets in detail. We use K-fold cross-validation on five datasets of *Cirrhosis*, *IBD*, *Obesity*, *T2D*, *WT2D*, and use Accuracy, Precision, Recall, F1-Score, AUC indicators to compare with previous methods in an all-round way. Figure 4 shows the visual comparison results of the four variants of the ablation experiment on five gut microbiome datasets. Each line represents a contour line, and the horizontal and vertical coordinates represent Accuracy and AUC metrics, respectively. In short, the method closer to the upper right region has better performance. The ablation experiments show that our method achieves the best results in almost all results, and the stability performs well, with minimal fluctuations.

Table 2: Contradistinction researchOur between our method and relatively good traditional machine learning algorithms on multiple indicators and stability analysis in the five OTU datasets.

| Metrics | Method | Cirrhosis | IBD | Obesity | T2D | WT2D |
|---------|--------|-----------|-----|---------|-----|------|
| Accuracy ↑ | RF | $0.877 \pm 0.043$ | $0.809 \pm 0.050$ | $0.644 \pm 0.028$ | $0.664 \pm 0.052$ | $0.703 \pm 0.105$ |
| | SVM | $0.834 \pm 0.052$ | $0.809 \pm 0.066$ | $0.636 \pm 0.042$ | $0.613 \pm 0.057$ | $0.596 \pm 0.102$ |
| | UMMAN(Ours) | $\mathbf{0.886 \pm 0.006}$ | $\mathbf{0.889 \pm 0.036}$ | $\mathbf{0.706 \pm 0.033}$ | $\mathbf{0.748 \pm 0.026}$ | $\mathbf{0.867 \pm 0.001}$ |
| Precision ↑ | RF | $0.890 \pm 0.041$ | $0.720 \pm 0.106$ | $0.540 \pm 0.109$ | $0.670 \pm 0.054$ | $0.730 \pm 0.114$ |
| | SVM | $0.840 \pm 0.052$ | $0.780 \pm 0.097$ | $0.560 \pm 0.103$ | $0.620 \pm 0.060$ | $0.590 \pm 0.132$ |
| | UMMAN(Ours) | $\mathbf{0.894 \pm 0.056}$ | $\mathbf{0.931 \pm 0.024}$ | $\mathbf{0.721 \pm 0.101}$ | $\mathbf{0.846 \pm 0.050}$ | $\mathbf{0.914 \pm 0.070}$ |
| Recall ↑ | Recall | $0.880 \pm 0.044$ | $0.810 \pm 0.050$ | $0.640 \pm 0.028$ | $0.660 \pm 0.052$ | $0.700 \pm 0.105$ |
| | SVM | $0.830 \pm 0.052$ | $0.810 \pm 0.066$ | $0.640 \pm 0.042$ | $0.610 \pm 0.057$ | $0.600 \pm 0.102$ |
| | UMMAN(Ours) | $\mathbf{0.952 \pm 0.001}$ | $\mathbf{0.969 \pm 0.023}$ | $\mathbf{0.758 \pm 0.048}$ | $\mathbf{0.973 \pm 0.022}$ | $\mathbf{0.857 \pm 0.001}$ |
| F1-Score ↑ | RF | $0.880 \pm 0.045$ | $0.750 \pm 0.073$ | $0.540 \pm 0.038$ | $0.660 \pm 0.053$ | $0.690 \pm 0.109$ |
| | SVM | $0.830 \pm 0.053$ | $0.780 \pm 0.076$ | $0.550 \pm 0.048$ | $0.6107 \pm 0.058$ | $0.570 \pm 0.112$ |
| | UMMAN(Ours) | $\mathbf{0.888 \pm 0.006}$ | $\mathbf{0.925 \pm 0.023}$ | $\mathbf{0.739 \pm 0.065}$ | $\mathbf{0.732 \pm 0.037}$ | $\mathbf{0.857 \pm 0.001}$ |
| AUC ↑ | RF | $0.945 \pm 0.036$ | $0.890 \pm 0.078$ | $0.655 \pm 0.079$ | $0.744 \pm 0.056$ | $0.762 \pm 0.111$ |
| | SVM | $0.922 \pm 0.041$ | $0.862 \pm 0.083$ | $0.648 \pm 0.071$ | $0.663 \pm 0.066$ | $0.664 \pm 0.126$ |
| | UMMAN(Ours) | $\mathbf{0.954 \pm 0.002}$ | $\mathbf{0.930 \pm 0.018}$ | $\mathbf{0.684 \pm 0.040}$ | $\mathbf{0.777 \pm 0.027}$ | $\mathbf{0.797 \pm 0.027}$ |

## 4.3 ABLATION STUDY

In our method, the Attention block and Node Feature Global Integration (NFGI) descriptor are two core components, introduced to improve the performance on the OTU datasets. We conduct an ablation study efficiency on four variants: a) Ours (Base), only with the framework of the UMMAN model; b) Ours (Attn), with the attention block which is used to get the embedding of the Original-Graph and the Shuffled-Graph. In this study, we use the average value of Multi-Graphs to replace in order to verify the efficiency of the Attention Block; c) Ours (NFGI), adopting the Node Feature Global Integration(NFGI) which is designed to describe the global graph embedding, replace this module with the average value of each node in this part; d) Ours (Full ), with both Attention Block and NFGI. The numeric comparisons on *Cirrhosis*, *IBD*, *Obesity*, *T2D*, *WT2D* are shown in Table 3. On the whole, the method with both Attention Block and DFGI, i.e., Ours (Full) performs the best. It is worth mentioning that even if our method is ablated, the effect on most indicators is better than previous work.

Table 3: Comparisons on the performance gains with the Attention Block and NFGI module in terms of two metrics.

| Method | ATTN | NFGI | Cirrhosis | | IBD | | Obesity | | T2D | | WT2D | |
|--------|------|------|-----------|------|-----|------|---------|------|-----|------|------|------|
| | | | Acc | AUC | Acc | AUC | Acc | AUC | Acc | AUC | Acc | AUC |
| Ours (Base) | ✗ | ✗ | 0.885 | 0.945 | 0.832 | 0.823 | 0.676 | 0.636 | 0.695 | 0.727 | 0.721 | 0.705 |
| Ours (Attn) | ✓ | ✗ | 0.882 | 0.933 | 0.818 | 0.844 | 0.657 | 0.676 | 0.685 | 0.715 | 0.733 | 0.773 |
| Ours (NFGI) | ✗ | ✓ | 0.861 | 0.929 | 0.808 | 0.774 | 0.672 | 0.637 | 0.721 | 0.770 | 0.800 | 0.767 |
| Ours (Full) | ✓ | ✓ | **0.886** | **0.954** | **0.889** | **0.930** | **0.706** | **0.684** | **0.748** | **0.777** | **0.867** | **0.797** |

## 5 CONCLUSION

In this paper, we propose a novel method UMMAN which combines graph neural network with intestinal flora disease prediction task for the first time in order to help learn the association between gut microbes and hosts. We first construct the Multi-Graph and Shuffled-Graph using multiple relation-types, and then update the embedding of nodes through Graph Convolutional Network. In addition, we introduce the Node Feature Global Integration (NFGI) to describe the embedding of the graph with node-level stage and graph-level stage. Finally, we design a joint loss consists of adversarial loss and hybrid attention loss as the final loss function. Extensive experiments show that our UMMAN performs well on the task of intestinal flora disease prediction and achieves the state-of-the-art.

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
