# OpenReview forum: "UMMAN: UNSUPERVISED MULTI-GRAPH MERGE ADVERSARIAL NETWORK FOR DISEASE PREDICTION BASED ON INTESTINAL FLORA"
_ICLR.cc/2024/Conference — Submitted to ICLR 2024_

### Official Review · Reviewer_QG5D · 2023-10-26

**Soundness:** 2 fair
**Presentation:** 2 fair
**Contribution:** 2 fair
**Rating:** 3
**Confidence:** 3

**Summary:**

This work aim to obtain the embeddings of nodes in the multigraph under unsupervised situation, so that it helps learn the multiplex relationship. The experiments are mainly conduct on disease network.

**Strengths:**

1. It learns multiplex connection between intestinal flora and hosts to guide the prediction of diseases.

2. This work proposes Nnde feature global integration descriptor to represent the global embedding of a graph.

3. The experiments of this work is good, comparing to existing algorithms.

**Weaknesses:**

1. The technique contribution of this work is not very high as it mainly uses the GNN for disease network.

2. The work may not related to ICLR community, as we can see that  most of reference are from bioinformatics.

3. The baselines are old, and the code/dataset are not available.

**Questions:**

1. The technique contribution of this work is not very high as it mainly uses the GNN for disease network.

2. The work may not related to ICLR community, as we can see that  most of reference are from bioinformatics.

3. The baselines are old, and the code/dataset are not available.

---

> ### Author Response · Authors · 2023-11-21
> **Reply to weakness and questions**
>
> “The technical contribution of this work is not very high as it mainly uses the GNN for disease network.” It may not sound very correct, as Amazon chief scientist Li Mu said: “A good paper does not have to appear all the time before. This is a module that no one has ever seen, but can you think of such a method?" First of all, as far as we know, previous GNN methods have never used multiplex relationships to construct graphs. At the same time, considering the particularity of this field, that is, the correlation of intestinal microorganisms is complicated and difficult to obtain, even multi-graphs cannot guarantee the accuracy of extraction. property, so we built Shuffled-Graph for adversarial training. At the same time, we supplemented ablation studies and tried it on Cirrhosis and T2D data sets. We found that if this method is not used, accuracy and auc metrics will drop by 2 ~6 percentage points, specific experimental data will be supplemented in the paper. In addition, “The work may not related to ICLR community”, in fact, the first topic in ICLR’s call for papers is “unsupervised, self-supervised, semi-supervised, and supervised representation learning”. Finally, "The baselines are old, and the code/dataset are not available.", you may not have read the full text carefully. First of all, there is very little related work in this field, and it is also one of the bottlenecks. Our method is the first time to apply GNN to Methods in this field, so previous work has never used GNN to solve this problem, and our supporting material contains the complete code, which is inconsistent with what you said.

---

### Official Review · Reviewer_hmSd · 2023-10-31

**Soundness:** 3 good
**Presentation:** 1 poor
**Contribution:** 2 fair
**Rating:** 5
**Confidence:** 4

**Summary:**

This paper firstly proposes a framework to model the gut microbiota disease prediction as a graph-level prediction task. The idea is to formulate multi-graphs to represent the multiplex connection between gut microbes and hosts, then a novel (problem-specific) graph learning architecture named UMMAN is proposed to generate the graph embedding. In UMMAN, GCN is utilized within each multi-graph to generate node embeddings and a novel readout function NFGI is proposed to generate the graph embedding corresponds to the multi-graph, then the attention mechanism is applied among multi-graphs. UMMAN is trained in an unsupervised manner.

**Strengths:**

Recently, graph neural networks (GNNs) have achieved impressive performance in various bioinformatical applications including drug discovery [1,2], Protein-Protein Interaction prediction (PPI) [3], molecular design [4], etc. This paper extends the applicability of GNNs to gut microbiota disease prediction and proposes a relevant unsupervised GNN named UMMAN. Experimental results demonstrate that UMMAN can beat previous machine learning models in the field, making it a pertinent and valuable research topic.

I would recommend the authors to add more discussion of the recent successes of GNNs in the context of bioinformatics in the introduction section to improve the manuscript.


[1] Wang, J.; Liu, X.; Shen, S.; Deng, L.; Liu, H. DeepDDS: deep graph neural network with attention mechanism to predict synergistic 601 drug combinations. Briefings in Bioinformatics 2022, 23, bbab390.

[2] Dong, Z.; Zhang, H.; Chen, Y.; Payne, P.R.O.; Li, F. Interpreting the Mechanism of Synergism for Drug Combinations Using Attention-Based Hierarchical Graph Pooling. Cancers 2023, 15, 4210. https://doi.org/10.3390/cancers15174210

[3] Zitnik, M., & Leskovec, J. (2017). Predicting multicellular function through multi-layer tissue networks. Bioinformatics, 33(14), 190-198.

[4] Olivecrona, M., Blaschke, T., Engkvist, O., & Chen, H. (2017). Molecular de-novo design through deep reinforcement learning. Journal of cheminformatics, 9(1), 48.

**Weaknesses:**

1. The presentation of the paper should be significantly improved. For instance,

   (1) In the last paragraph in page 5, 'On the wholethe function $\mathcal{G}$ denotes the embedding of the graph ...', delete 'the'.

   (2) In the same paragraph, 'Let {$x^{∗}_{j}$} _{j=1,2..,n} be the each center of the histogram bins', replace n with K (number of bins)

   (3) The description of 'Graph-Level stage' in NFGI is unclear.

   (4) In function 11, $\mathcal{T}$ and $H^{(t)}$ are not clearly defined.

   (5) The definitions of Original-Graph and Shuffled-Graph is not very clear. It is better to provide formal definitions.

   (6) ...

2. The objective of the design of NFGI is unclear. Since the sum of node embeddings is a widely used graph-level readout function, both the node level stage/descriptor (NLD) and graph level stage/descriptor (GLD) are used to summarize extracted node embeddings to a graph-level vectorial representation.

3. The experiment section is not very sound. Recent SOTA (state-of-the-art) expressive GNN baselines are not included. Since the gut microbiota disease prediction is studied as a graph prediction task, these powerful GNN baselines are necessary.

4. The paper concludes that the proposed UMMAN is stable. The experimental section does not support it.

**Questions:**

1. It is not clear why the unsupervised training is used. What's the advantage over supervised learning on the gut microbiota disease prediction?

2. Nodes in a graph can be randomly permuted, thus these there is no fixed order of nodes in a graph. Then what is the meanings of 'position of edges' and 'disrupt the nodes' in the sentence 'we design the adversarial control group, i.e., keep the position of the edges unchanged, randomly disrupt the nodes to train part of the discriminator'? Does that mean we keep the adjacency matrix yet permute the node order?

---

> ### Author Response · Authors · 2023-11-21
> **reply to weakness and questions**
>
> We sincerely appreciate your suggestions for improving the details of the paper. Our method is the first to apply GNN to this field, so previous work has never used GNN to solve this problem. In view of this, we used GraphSAGE as a control experiment in the comparison experiment in the original paper, as shown in table1 . In the past two days, we have also added GAT experiments, and the results are similar to the GraphSAGE in the table. The specific data will also be displayed in the final paper. In addition, previous methods to extract the global features of a graph usually use the average of all nodes or global pooling, but the method is simple and the effect is not ideal. In order to verify the stability of the NFGI module, we conducted experiments on the value of K and found that the model with K ranging from 10 to 20 is generally more stable. Thank you again for your suggestions for improving the details of the paper. We will continue to revise and polish it.

---

> > ### Author Response · Authors · 2023-11-21
> > **Another reply to weakness and questions**
> >
> > "It is not clear why the unsupervised training is used." Because the hosts in the field of bioinformatics are usually patients or related personnel, and because the label involves user privacy, unsupervised learning is a very important research direction in bioinformatics. We have never used the host's label in the process of obtaining the embedding of each node, so this is our explanation for this problem.

---

> > > ### Comment · Reviewer_hmSd · 2023-11-22
> > >
> > > Thank you for your response, yet the weaknesses are not well-addressed in the response.

---

### Official Review · Reviewer_uFaR · 2023-11-01

**Soundness:** 2 fair
**Presentation:** 2 fair
**Contribution:** 2 fair
**Rating:** 3
**Confidence:** 3

**Summary:**

In this paper, the authors study for a practical problem, the intestinal flora disease prediction task, which predict disease through abundance information of gut microbes.

The major challenge of this task is that there is a multiplex and implicit connection between gut microbes and hosts, which makes the task difficult if only using the abundance information. In light of this challenge, the authors propose a method UMMAN which combines graph neural network to help learn the association between gut microbes and hosts.

Original-Graphs, are constructed by using multiple relation-types, and then utilized to update the embedding of nodes through Graph Convolutional Networks. Attention mechanism is also introduced to update the embeddings of the corresponding nodes of the Multi-Graphs.

Then the relationship among nodes of the Original-Graph is destroyed to get the corresponding Shuffled-Graph, which will be used for adversarial learning to obtain the correlation among nodes more reliably. The authors design the adversarial control group, i.e., keep the position of the edges unchanged, and randomly disrupt the nodes to train part of the discriminator at the same time.

In addition, the authors introduce the Node Feature Global Integration (NFGI) to describe a more comprehensive embedding of the graph with node-level stage and graph-level stage.

**Strengths:**

The idea of introducing GNN for the intestinal flora disease prediction task by learning the inner association between gut microbes and hosts is new to the field.

**Weaknesses:**

1. Technical contribution is a bit marginal as both the applied model and the idea of contrast learning are somehow take-off-shelf.

2. The motivation for applying GNN needs to be clearly explained i.e. why the columns in the tabular dataset are transformed into the nodes on a graph? Specifically, it is not very clear why some microbes are considered to be more correlated with each other and that they should be connected by an edge on a graph. The paper only provides a simple claim that two nodes are considered to be connected if the distance between their embeddings is below a variable threshold. It would be better if the authors could provide more details to illustrate the motivation. Moreover, as introducing graph machine learning to the field is one of the main contributions, it would be better if the authors could provide more details and instructions on the embedding process. For example, what are the features taken into consideration, the abundance level, the microbes' class, or anything else?

3. The motivations of the proposed method (especially for Shffuled-Graph and NFGI) need to be clearly and explicitly explained. For example, it is not well motivated why we need to perturb the original graph by exchanging nodes.

For the claim that ‘basic CNNs can't beat the traditional machine learning algorithm in this case (OTU datasets that are extended to table types) because the basic CNN’s kernel cannot learn arbitrary transformed features, which is not suitable for OTU datasets,’ it would be better for the authors to provide more details about the logic chain and explain what is meant by "arbitrary transformed features" in the context of OTU datasets.
Similarly, it would be better to tell more about why exchanging the column order in the tabular dataset will not affect the final result.In the introduction, it is mentioned that 'The exchange of rows and columns of the datasets do not affect the results of pattern recognition.'The authors should explain why this property is not desired i.e., why the authors desire to affect the results of pattern recognition by exchanging the rows and columns of the datasets and how it motivated them to propose their method.

For the proposed Shuffled-Graph, the motivation should also be explicitly illustrated. E.g., why some microbes are considered to be more correlated with each other, and what is the benefit of perturbing this kind of correlation? Why the Shuffled-Graph generated by breaking the correlation among nodes can be used as the negative adversarial control group, and why does such an adversarial learning method work?  It would be better for the authors to add proper references or experiments to support the claim that such perturbation should work.
Moreover, for NFGI, why such a component is desired, and why do the authors design such a proposed graph-level stage instead of other graph-level operations such as pooling?

4. It would be better if the authors could provide more details of the proposed methods, especially those of core contributions. E.g., how the K bins are chosen and decided for the GLD in NFGI.

5. What is the 'true embedding', and how can it 'includes the complex and implicit relationship between gut microbes and hosts'?

**Questions:**

1. How the existing works conducted on the tabular data can be introduced in the Related Works with more details. E.g., what a typical tabular data look like? A figure with a few columns and rows from the tabular data should be enough

2. The motivations of any proposed method should be clearly and explicitly explained instead of just telling the reader how or what to do with the proposed method. Even some toy examples can provide a more comprehensive introduction for readers.

3. Some notations appear in the equations for the first time and are not mentioned in the text. It would be better if the authors could align them with the text, e.g., H(which should align with ‘global embedding obtained by NFGI’) in eq11 and P(which should align with ‘global embedding matrix of real graph’) in eq12.

---

> ### Author Response · Authors · 2023-11-21
> **reply to weakness and questions**
>
> I have carefully read all your suggestions and agree with most of the opinions. As you said, we should introduce the form of tabular data and the shortcomings of basic CNNs in more detail in the paper, and will continue to supplement the paper in these aspects. However, "technical contribution is a bit marginal" may not be very reasonable, because as Amazon chief scientist Li Mu said: "A good paper is not about appearing modules that no one has seen before, but about such methods. Can you think of it?" First of all, as far as we know, previous GNN methods have never used multiplex relationships to construct graphs. At the same time, considering the particularity of this field, that is, the correlation of intestinal microorganisms is complicated and difficult to obtain, even multi-graphs cannot guarantee the accuracy of extraction. property, so we built Shuffled-Graph for adversarial training. At the same time, we supplemented ablation studies and tried it on Cirrhosis and T2D data sets. We found that if this method is not used, accuracy and auc metrics will drop by 2 ~6 percentage points, specific experimental data will be supplemented in the paper.

---

> > ### Comment · Reviewer_uFaR · 2023-11-23
> > **Response to Authors' Rebuttal**
> >
> > Thank you for your response.  However, the weaknesses are not well-addressed in this response. Hence, I maintain the original score.

---

### Official Review · Reviewer_9ne7 · 2023-11-01

**Soundness:** 3 good
**Presentation:** 3 good
**Contribution:** 2 fair
**Rating:** 5
**Confidence:** 3

**Summary:**

The paper proposes a new unsupervised graph neural network architecture called UMMAN for predicting diseases based on intestinal flora data. The key ideas are: 1) Construct multi-graph representations of the flora-host relationships using different similarity metrics. 2) Introduce an adversarial training scheme that makes the model embeddings agree with the true graph while disagreeing with a shuffled graph. 3) Propose a two-stage Node Feature Global Integration module to characterize both local node and global graph features. Experiments show state-of-the-art performance on multiple disease datasets compared to previous methods.

**Strengths:**

- Novel application of graph neural networks to microbiome disease prediction, allowing the model to learn relevant flora-host relationships.
- Interesting adversarial training approach with shuffled graphs as a regularization method.
- Thorough experiments demonstrating superior performance over existing techniques on multiple datasets.

**Weaknesses:**

- The proposed adversarial training scheme seems a bit ad-hoc. More analysis could be provided on why this particular approach is effective.
- Ablation studies only evaluate the contribution of individual components; would be good to see the ablation of a full adversarial training scheme.
- More discussion could be provided on the choice of graph construction methods.

**Questions:**

- How dependent is the performance on the specific graph construction techniques used? Have other graph representations been explored?
- Is the adversarial training approach specifically tailored for this problem setting, or does it represent a more general regularization technique?

---

> ### Author Response · Authors · 2023-11-20
> **reply to weakness and questions**
>
> Thank you for your suggestion. First, for "adversarial training", we supplemented ablation studies and tried it on Cirrhosis and T2D data sets. We found that if this method is not used, accuracy and auc metrics will drop by 2 to 6 percentage points. Specific experimental data It will be supplemented in the paper. As for the motivation of Shuffled-Graph, the paper actually involves it but does not elaborate on it. Specifically, the connection between intestinal microorganisms and the host is intricate. Even if we use multiple relationships to build graphs, it is difficult to ensure the accuracy of the correlation acquisition. This is also one of the biggest pain points in this field, so we thought of another way. , that is, constructing Shuffled-Graph. During training, the model expects the difference between Shuffled-Graph and Original-Graph to be maximized, that is, to maximize the similarity between Original-Graph and true embedding. In addition, our method is the first time to introduce GNN in this field. Although there have been many GNN methods before, such as STGCN, GAT, and GraphSAGE, among which GraphSAGE is also used as part of the experiment, it is not applicable in this field.

---

### Meta-Review · Area_Chair_Momu · 2023-12-07

**Metareview:**

This paper proposes a framework to model the gut microbiota disease prediction task as a graph-level prediction problem. The framework represents the multiplex connection between gut microbes and hosts as multi-graphs, and then adopts a novel graph learning architecture named  Unsupervised Multi-graph Merge Adversarial Network (UMMAN) to get the graph embedding. UMMAN introduces an adversarial training scheme, and proposes a two-stage Node Feature Global Integration module, trains the model in an unsupervised manner.

The application is  novel for GNN. The method introduces some interesting modules, such as the Node Feature Global Integration (NFGI) module. The experiments show the method achieved somehow satisfactory performance.

The novelty is limited, since the model and contrast learning is not new. More experiments should be provided, such as ablation study, using more powerful GNNs. The paper should be carefully polished, to make the motivations and details more clear.

**Justification For Why Not Higher Score:**

The contribution is marginal, and more experiments should be provided.

**Justification For Why Not Lower Score:**

N/A

---

### Decision · Program_Chairs · 2024-01-16

Reject